# Drug-Eluting Sutures by Hot-Melt Extrusion: Current Trends and Future Potentials

**DOI:** 10.3390/ma16227245

**Published:** 2023-11-20

**Authors:** Garba M. Khalid, Nashiru Billa

**Affiliations:** 1Department of Pharmaceutics, UCL School of Pharmacy, University College London, 29-39 Brunswick Square, London WC1N 1AX, UK; khalid.mohammed@ucl.ac.uk; 2FabRx Ltd., Henwood House, Henwood, Asford TN24 8DH, UK; 3Pharmaceutical Sciences Department, College of Pharmacy, QU Health, Qatar University, Doha P.O. Box 2713, Qatar

**Keywords:** suture, drug-eluting suture, hot-melt extrusion, polymer suture, production method, surgical site infection, wound healing

## Abstract

Surgical site infections (SSIs) may result from surgical procedures requiring a secondary administration of drugs at site or systemically in treating the infection. Drug-eluting sutures containing antimicrobial agents symbolise a latent strategy that precludes a secondary drug administration. It also offers the possibility of delivering a myriad of therapeutic agents to a localised wound site to effect analgesia, anti-inflammation, or the deployment of proteins useful for wound healing. Further, the use of biodegradable drug-eluting sutures eliminates the need for implanting foreign material into the wound, which needs to be removed after healing. In this review, we expound on recent trends in the manufacture of drug-eluting sutures with a focus on the hot-melt extrusion (HME) technique. HME provides a solvent-free, continuous one-step manufacturing conduit for drug-eluting sutures, hence, there is no drying step, which can be detrimental to the drug or suture threads and, thus, environmentally friendly. There is the possibility of combining the technology with additive manufacturing platforms to generate personalised drug-loaded implantable devices through prototyping and scalability. The review also highlights key material requirements for fabricating drug-eluting sutures by HME, as well as quality attributes. Finally, a preview of emerging drug-eluting sutures and advocacy for harmonisation of quality assurance by regulatory authorities that permits quality evaluation of novelty sutures is presented.

## 1. Introduction

Sutures are biomedical devices comprising fibre(s) that hold surrounding tissues of a wound together, seal surgery sites, or squeeze blood vessels to achieve haemostasis [1,2]. Sutures have long been used to restore injured tissues to their original structure and functions [3]. Surgical sutures assist in the closure and healing of trauma-induced and surgical wounds by upholding adjoining tissues together to facilitate the healing process [4].

Based on biodegradability, sutures are generally classified as absorbable or non-absorbable, depending on whether they degrade or not post-application to the wound. Absorbable sutures degrade by hydrolysis or proteolytically shortly after application to the wound. The US Pharmacopeia defines absorbable sutures as biomaterials that “lose most of their tensile strength within 60 days post-implantation in vivo” [5,6]. Non-absorbable sutures offer long-term mechanical support to the tissue and can be removed only by external means [5]. Non-biodegradable sutures have to be removed post-healing of a wound. However, removal of sutures is clinically challenging, particularly in difficult-to-access anatomical areas or in paediatric patients. In such cases, using biodegradable sutures is preferred for the safety of the patient and the success of the treatment [7,8]. The two classes can be further categorised as natural or synthetic based on the materials they are made from. For example, catgut, which is derived from sheep or goat intestines and contains over 99% pure collagen, was the first available naturally absorbable suture and is broken down through a combined action of proteolytic enzyme degradation and phagocytosis within 7–10 days [9]. Historically, catgut suture has been commercially available in two forms: plain catgut introduced in 1860 and treated catgut tanned by chromium trioxide (chromic catgut) introduced in 1880, which has stronger mechanical properties, slower degradation rate, and lower tissue reactivity compared to plain catgut [9,10]. Polyglycolic acid, on the other hand, introduced in early 1970, was the first synthetic absorbable suture with higher tensile strength. It is absorbed within 60–90 days by hydrolysis in the human body [11]. The most frequently used natural non-absorbable sutures include silk, cotton, and linen. Silk has uniquely excellent handling and knot-tying properties but possesses high tissue reactivity and can be used for cardiovascular, ophthalmic, and neurological procedures [12,13]. Though considered non-absorbable, silk degrades slowly by proteolytic degradation and is absorbed slowly over 1–2 years [14]. Nylon is a synthetic polyamide non-absorbable suture with a major drawback of having a great deal of memory, which is referred to as the inherent ability of the suture material to return to or maintain its original shape [5,15]. Other synthetic non-absorbable sutures are polypropylene sutures known to maintain excellent tensile strength with a low coefficient of friction, giving less tissue trauma in suturing and low tissue reactivity (Calhoun and Kitten, 1986). Polypropylene sutures have been used in plastic, cardiovascular, orthopaedic, and ocular surgery [16,17].

Structurally, however, there are three different sutures, namely, (i) monofilament sutures, (ii) multifilament (braided) sutures, and (iii) barbed sutures. A monofilament suture consists of only a single strand of fibre, whereas multifilament sutures are composed of several strands braided together. Braided sutures exhibit more flexibility and better tensile strength than non-braided ones. However, with the former, there is less tissue reaction and scar formation after wound healing due to the high capillary network in the braided sutures, which has also been associated with favouring bacterial proliferation [18,19]. Barbed sutures, on the other hand, are knotless sutures containing specially engineered barbs that invade the tissues and keep them in place. Barbed sutures are available in both absorbable and non-absorbable categories. They are monofilament in nature, and the barbs serve to grip the sutured tissue in a continuous manner and retain tensile strength [20]. The main benefits of barbed sutures include the elimination of surgical knots, knot-related complications, and increased efficiency of wound closure [5,21]. The size and spacing of the barbs, which are integrally formed into the core, are designed to provide maximum holding in soft tissue such as fascia and provide tactile feedback to regulate tension. They are also popular for abdominal wall repair following free flap harvest [19,22].

Surgical care may be required in a broad range of medical conditions in patients [23] of all ages. Furthermore, surgery can be curative, as in many cancer cases [24], or preventative, as in prophylactic mastectomy [25]. It is often a component of acute emergency care, such as wound care/suturing after accidents, bowel perforations, and trauma, as well as the treatment of chronic diseases, such as osteoarthritis and inflammatory bowel disorders. Thus, surgical care is a fundamental component of healthcare and contributes to overall social and economic development [24,26,27]. Therefore, in one way or another, these surgical procedures require the use of medical devices, including surgical sutures. Of interest to this review is a recent survey, which highlights an increase in the total volume of global surgical procedures by 5.1% from 2021 to 2025 [28]. This is driven by ageing populations and increased incidence of diseases caused by poor lifestyle choices. Hence, the global surgical consumables market is estimated to reach USD 19.51 billion by 2025, with surgical sutures accounting for approximately 19% of the total market by value, equivalent to about USD 4.60 billion [28]. In addition, demands for suture materials are on the rise due to an increased number of surgical procedures performed worldwide, regardless of the availability of suture substitutes, such as surgical staples, glues, and strips, in the market. These products often fall short of the stability and flexibility rendered by sutures in wound management, especially the advanced types of sutures with enhanced functionalities, such as drug-eluting sutures [29]. Because of their versatility in wound closure and/or immobilising prosthesis, suture threads are the most widely used implants in surgery. Currently, there are hundreds of different sutures made of synthetic or natural polymers commercialised, that vary in chemical composition, mechanical properties, suture structure (monofilament, braided, or barbed), knot security, diameter, and absorption rate [2]. Aptly, databases (Embase^®^, PubMed^®^, and Scopus^®^) present about a three-fold increase in research output related to surgical sutures in the last two decades, related to the search for new materials, development of drug-eluting surgical sutures with enhanced functionalities, such as barbed sutures and stretchable sutures, able to return to their original shape after application. Smart sutures are coated with growth factors [30,31,32,33] or stem cells [34,35,36].

Drug-eluting sutures provide an opportunity to facilitate wound/surgery healing and avoid repeated surgery, improving patient compliance and overall therapeutic success. These newly designed sutures represent a timely response to critical market needs due to the increased potential for bacterial biofilm growth that causes surgical site infections (SSI) and undesirable inflammatory responses when traditional surgical sutures are used [37]. Hot-melt extrusion of material may provide a platform for the efficient production of drug delivery systems, such as drug-eluting stents. However, a literature search using a combination of the keywords ‘drug-eluting suture’ and ‘hot-melt extrusion’ from the above databases showed very scanty outputs. This highlights a budding opportunity for research and development in drug-eluting sutures prepared by hot-melt extrusion technology.

Thus, in this review, we aim to provide an appraisal of the recent developments in drug-eluting surgical sutures and trends in preparation techniques with a focus on hot-melt extrusion (HME). Furthermore, we discuss the material requirement for fabricating drug-eluting sutures by HME, critical quality attributes, and their assessment protocols. We then conclude with a look at the future perspectives on drug-eluting surgical sutures by HME prototyping through optimised product design, preparation, characterisation, and scalability. Single or combination of the following keywords: ‘surgical suture’, ‘drug-eluting suture’, ‘hot-melt extrusion’, ‘polymer suture’, ‘production method’, ‘surgical site infection’, ‘wound healing’ were used to search the literature from Embase^®^, PubMed^®^, Scopus^®^, and Web of Science^®^ databases.

## 2. Drug-Eluting Surgical Sutures

The materials and methods should be described with sufficient details to allow others to reproduce surgical sutures xenogenic to the body, and as such, cause tissue responses [5]. Indeed, implantation of sutures inside the human body may trigger inflammatory responses and/or postoperative surgical site infection (SSI), requiring additional interventions. These additional responses are usually therapeutic to (i) reduce the inflammatory response and the subsequent pain and/or (ii) to prevent infections associated with implanted medical devices, such as postoperative surgical site infection, which is difficult to cure once it happens [2]. This may often prolong hospital stays and require multiple surgeries and high-dose drug administration due to low bioavailability and non-specificity to target [2,38]. Such non-site-specific drug administration may further exacerbate side effects. Hence, localised drug delivery at the injury site and/or the surgery site, along with the sutures, is a rational approach [38].

Drug-eluting sutures represent the next generation of surgical sutures since, in addition to fulfilling their mechanical functions during wound healing, they also deliver the drug to the wound. Incorporating active pharmaceutical ingredients (APIs) into the suture transforms it into a localised drug delivery system that minimises systemic side effects associated with postoperative surgical procedures. Ultimately, patient compliance and treatment success are assured with drug-eluting sutures. The final product is considered a ‘combination product’ by the US Food and Drug Administration (FDA) [39]. The application of biodegradable sutures to deliver drugs not only eliminates the need for implanting foreign material at the wound site but also effectively reduces the possibility of SSI, inflammatory responses, and pains at surgery sites [3]. In addition, it is possible to deliver therapeutic proteins to aid in repairing damaged tissues [40]. The incorporation of sutures with therapeutic agents may generate a medical device that can control and optimise drug release at the wound site [38]. Sustained release of drugs at a specific surgery site can allow therapeutically relevant concentration locally for a prolonged duration without exceeding the toxic limit in the systemic circulation. The challenge in fabricating a drug-eluting suture is to obtain the required concentration and potency of the API without compromising on the mechanical properties of the suture and this can be achieved by incorporating polymer degradation and controlled drug release capabilities in the suture [39].

### 2.1. Unique Advantages of Drug-Eluting Sutures

As the name implies, drug-eluting sutures exhibit some unique benefits over conventional drug-free sutures, some of which are enumerated below [4,40,41,42,43];

Localised discharge of antimicrobial agents, anti-inflammatory drugs, analgesics, local anaesthetics, extracellular matrix proteins, and cytokines to the wound site.Eliminate the toxicity and/or side effects associated with peroral and/or parenteral drug administration.Reduce or eliminate the development of surgical site infections.A combination of multiple drugs, e.g., antimicrobial agents and anti-inflammatory drugs in the same suture, can exhibit synergistic effects and/or additive effects on the affected site.Better therapeutic concentration with prolonged duration can be achieved with amplified loading of drug and sustained drug delivery.Better and faster wound healing and tissue regeneration.Enhanced mechanical properties, especially when the loaded drug(s) act as a filler to reinforce the tensile strength of the suture threads.Applicable for use in invasive surgeries, regenerative medicine, and tissue engineering.

The first US Food and Drug Administration (FDA) approved drug-eluting suture was Vicryl Plus, granted in 2002, and commercialised by Ethicon Inc. [4]. Vicryl Plus comprised of polyglactin coated with antibacterial triclosan. Following this approval, triclosan-coated sutures are now widely used and serve to overcome bacterial adherence to the wound [44,45]. Sutures containing bioactive substances can also be used for different site-specific procedures. These advances have propelled drug-eluting sutures to be good candidates not only in invasive surgeries and regenerative medicine but also in tissue engineering [40]. Kashiwabuchi and co-workers developed absorbable antibiotic-eluting sutures of levofloxacin for ophthalmic surgery by wet electrospinning, which provides suitable suture size requirements and sustained drug release [42]. Indeed, the antimicrobial octenidine was tested as a drug-eluting surgical suture by coating it onto a polyglycolic acid suture as a fatty acid of either octenidine-laurate or octenidine-palmitate with a high antimicrobial effect and biocompatibility [46]. In addition, a curious finding from Weldon’s research team demonstrates the feasibility of fabricating a bupivacaine (local anaesthetic) eluting suture which combines the function and ubiquity of the suture for wound repair with the controlled release properties of a biodegradable polymeric matrix poly(lactic-co-glycolic acid) (PLGA) by electrospinning [47]. To further highlight the additional unique nature of drug-eluting suture systems, the work of Haley and colleagues demonstrates the dual loading of both anti-inflammatory and antimicrobial drugs onto a polymerised cyclodextrin-coated surgical suture capable of locally delivering the drugs throughout the phases of acute and chronic wound healing. They succeeded in loading rifampicin and resveratrol by surface coating [37]. Therefore, research and application of drug-eluting sutures in modern-day science and clinical practice are just setting the phase in light of emerging trends in the production and utilisation of these medical devices.

### 2.2. Production Methods of Drug-Eluting Sutures

Drug-eluting sutures are produced by different manufacturing processes. Currently, two general approaches are utilised in their fabrication: (i) addition of the API to the material during the manufacturing of the suture and (ii) loading the API to an already manufactured suture [2]. For the latter, the drug-loading can be achieved in two ways: drug loading onto an already manufactured suture by coating, soaking, or grafting, and drug loading into an already manufactured suture (dispersion) through soaking or supercritical CO_2_-assisted impregnation. While API incorporation during suture manufacturing can be achieved through spinning (uniaxial spinning), electrospinning (coaxial), or hot melt extrusion processes [4,48]. Electrospinning is used to achieve porous nanofiber sutures [49]. The drug/polymer blend is injected into the syringe and electrospun into nanofibers. This technique can fabricate suture materials into porous nanofibers with diameters in the range of 5 to 500 nm [2]. Each of the mentioned production methods has merits and some drawbacks. For instance, electrospun nanofibers generally have poor mechanical performance since they are in nonwoven form. Even though not a rule of thumb, in most cases, the thinner the fibre diameter of electrospun sutures, the higher the mechanical strength [50,51]. Indeed, a literature search from the Scopus database with ‘drug loaded surgical sutures by hot melt extrusion’ and by other surgical sutures fabrication techniques, as listed in Figure 1 as of November 4th 2023, returned very few research articles on suture preparation by hot melt extrusion, a total of 38 (1.5% relative to other preparation methods) over the last ten years (2013 to 2023). The most frequently reported technique is grafting (42.0%), followed by the coating technique (32.7%), then electrospinning (15.0%). This shows a significant paucity of research output on drug-eluting sutures by hot melt extrusion despite its promising potential. Therefore, in this review, we focused on hot melt extrusion due to its emerging potential as a method of choice for ease of processability and scalability. It is also the opinion of the authors that, through hot-melt extrusion, drug-eluting sutures with enhanced and/or modified drug release properties can be fabricated, especially for poorly soluble class II and IV Biopharmaceutic Classification System (BCS) drugs due to the possibility of generating amorphous solid dispersion of such drugs with enhanced solubility and drug release profiles.

### 2.3. Production of Drug-Eluting Sutures by Hot-Melt Extrusion

Hot-melt extrusion (HME) offers a one-step, easy-to-implement, solvent-free, and continuous manufacturing modality for drug-eluting sutures [2,52,53]. Preliminarily, the polymer is blended with the API(s) and other excipients and often sifted through a sieve of appropriate nominal size. The mixture is then fed into the hot-melt extruder and melted together with the API and other excipients and extruded through a die of a defined diameter instead of being solubilised in a solvent (Figure 2). The continuous use temperature (CUT) during the extrusion is selected based on the physicochemical properties of the API and that of the thermoresistant polymer. This eliminates the solvent effects on the API, the polymer, and the potential toxic effect of the residual solvent on the patients. In this process, the API is homogeneously distributed within the suture, which is crucial for reproducibility. Moreover, to control the drug release rate from the suture, the API can also be modified prior to mixing with the polymer [4,38,54]. The suture threads obtained by HME have a diameter ranging from 50 to 300 μm, hence, monofilament in nature [2].

There are few studies that have reported the manufacturing of drug-eluting sutures by HME [55,56]. This is largely due to the elevated processing temperature required for the extrusion. Thus, it cannot be applied to thermolabile drugs [1]. Polycaprolactone has received much attention for use in this regard because it is biodegradable and is approved by the FDA for human use [38,56,57].

HME has been widely used in pharmaceutical engineering as a solvent-free drug delivery manufacturing technology not only in the production of drug-eluting sutures but also to fabricate personalised orodispersible films that ease the oral administration of drugs to patients with dysphagia or limited access to and/or intake of water [58,59,60]. In addition, it is used in cutaneous patches loaded with drugs [61], extended-release tablets matrix [62], chronotopic tablets for pulsatile and/or colonic delivery of drugs [63], and the manufacture of fixed-dose combination oral drug delivery system containing two different drugs exhibiting immediate and delayed-release in the management of the cardiovascular disease [64,65]. Thanks to its ability to combine poorly water-soluble drugs in polymers it is used for enhanced drug solubility, dissolution rates, and drug release [59,66,67]. For instance, Nagy and co-workers developed a novel solvent-free melt electrospinning method to prepare drug-loaded amorphous Eudragit^®^ E sutures with an ultrafast release of poorly water-soluble drug carvedilol. The method involves the preparation of Eudragit^®^ E (80%) and carvedilol (20%) blend by hot-melt extrusion at 130 °C, followed by melt electrospinning of the final suture filament. The advantages of this approach include a solvent-free, continuous process, and effective amorphisation of the drug to enhance its solubility and release [55]. In another related scenario, Deng and co-workers proposed absorbable composite drug-eluting sutures for wound healing containing an anti-inflammatory drug, diclofenac potassium, by hot-melt extrusion. The suture materials were composed of a medical-grade poly(ε-caprolactone), poly(ethylene glycol), hydrolysed keratin powder, and chitosan, which resulted in a suture with homogeneous physicochemical properties, robust mechanical properties, cell-friendly biological properties, and tuneable drug release at the wound site [38]. These and other practical examples of drug-eluting sutures with unique features fabricated by HME are summarised in Table 1.

### 2.4. Advantages of Hot-Melt Extrusion in Drug-Eluting Suture Fabrication

Hot-melt extrusion has several advantages over other solvent-based drug-eluting suture production methods, as enumerated below [38,49,57,66]:HME can disperse a single or combination of different APIs in the polymer matrix at the molecular level with the likelihood to enhance the dissolution of poorly soluble drugs when they form amorphous solid dispersion of the drug–polymer blend.APIs can be homogeneously distributed into the suture cross-section. Hence, achieving prolonged drug release.It is a one-step continuous process, hence, can be easily scaled up, making it easy to translate from laboratory to commercial production.There is no need to dissolve the suture-forming polymer prior to drug-loading.The process does not require any solvent recovery, which can be costly.HME process is devoid of risk of solvent explosion or additional drying procedure that may be detrimental to the drug and/or drug–polymer mix.There is no residual solvent in the suture, therefore, less undesired and/or side effects to the application/wound site.A quantitative yield of 100% is achievable.HME is a safer, environmentally friendly, and ‘greener’ technology for thermostable drugs than organic-solvent-based techniques for drug-eluting suture preparation. Hence, it is a cheaper technique.HME can be coupled with melt electrospinning, and thus, continuous manufacturing is feasible [49].Melt-spun sutured filaments are usually void-free, with diameters and tensile strengths within the USP limits [70].

These merits have the potential to increase the applicability of HME as a method of choice for preparing drug-eluting sutures in the pharmaceutical industry. Furthermore, considering the current trends in digital health technology and advancements in pharmaceutical engineering and drug delivery technologies, HME can be coupled with additive manufacturing (AM), particularly three-dimensional (3D) printers, to generate personalised drug-eluting sutures with customisable drug-loading and/or drug release patterns based on individual patient conditions or surgical procedure required.

Despite these advantages and potential use as a continuous process, only a few studies have reported the use of HME for the manufacturing of drug-eluting sutures [56]. This could be explained by the need for high processing temperatures in some cases that may not be favourable to some APIs. Indeed, this process cannot be applied to thermolabile APIs with polymers that exhibit high melting temperatures [2], as discussed below.

### 2.5. Limitations of HME for the Production of Drug-Eluting Sutures on Drug Half-Life

Among the most critical limitations of HME is the thermal degradations of the drug and/or drug–polymer mix. HME possesses intrinsic limitations with respect to processing temperature on thermolabile active pharmaceutical ingredients on the drug half-life due to the application of high thermal and mechanical energy during extrusion [71]. However, a study by Deng et al. indicates that mixing an appropriate amount of drug and polymer(s) can result in a blend with suitable thermal stability [38]. For instance, with the addition of a 4% weight of chitosan/keratin blend into poly(ε-caprolactone) and poly(ethylene glycol) blend, the thermal stability of the suture fibres loaded with diclofenac potassium was maintained at a degradation temperature of 378 °C similar to pure poly(ε-caprolactone) of 375 °C thanks to the solvent-free manufacturing process by HME. Similarly, sometimes, the degradation products from the HME process need to be investigated and monitored to ensure their limits do not exceed the recommended amount in pharmaceutical products. As an example, Khalid and colleagues found an impurity in the diclofenac sodium as a degradation product from orodispersible films manufactured by HME 3D printing due to thermal exposure of the drug. The detected amount monitored under a 3-month accelerated stability study was found to be below the acceptable limits set by the European Pharmacopoeia (Ph. Eur.) [55]. Thus, in a broader perspective, the detection of drug degradation products does not mean a complete stoppage of utilising the HME process, but additional quantification and follow-up analyses are needed to justify the adoption of HME or otherwise. Therefore, as a rule of thumb, an initial assessment must be conducted on individual products and the drug–polymer mix during the pre-formulation phase. Traditionally, chromatographic techniques, such as high-pressure liquid chromatography (HPLC), are used to evaluate the chemical stability of pharmaceutical actives ingredients (API) subjected to elevated temperature and shear forces during HME. HPLC equipped with sensitive detectors, such as mass spectrometry (MS), enables the identification of degradation products [72]. Other detectors linked with HPLC may also be used to assess degradation profiles from HME products, including HPLC with Photo Diode Array detectors (HPLC-PDA) and Gas-Chromatography with Mass Spectrometry detectors (GC-MS) [71,73].

In addition to chromatographic techniques used for the identification and quantification of API and degradation products from HME, evaluation of the physical state is also crucial in relation to the stability of the extrudate since the heat generated during HME often disrupts the crystal lattice of the API and the transformation to amorphous or partially amorphous configuration. This subsequently affects the stability and shelf life [71]. Various techniques are available for assessing the physical conformation of drug–polymer systems subjected to HME, including X-ray powder diffraction (XRPD), differential scanning calorimetry (DSC), temperature modulated DSC (TMDSC), hot stage microscopy or hot stage polarised light microscopy, and atomic force microscopy [71,74].

Other constraints of HME are related to the limited number of heat-stable polymers, the requirement of raw materials with high flow properties, limited physical stability, and often the precipitation of drugs during dissolution [73]. The later limitations aside thermal degradation potentials can be circumvented by utilizing drug–polymer combinations that manifest stable interactions, such as hydrogen bonds, dipole–dipole, and van der Waals interactions [73].

Moreover, with HME, an increase in the drug content has been shown to decrease the mechanical properties of the suture [10]. These are some of the drawbacks of this approach. Building upon this premise and to the best of our knowledge, commercially available drug-eluting sutures manufactured by HME or patented products in this category are very scarce. Most of the commercially available drug-loaded sutures are based on coating techniques [70]. Therefore, the limited number of drug-eluting sutures commercially available indicates considerable scope for research and product development, especially using HME as a cost-effective and easy-to-use continuous manufacturing technique. Table 1 exemplifies some biodegradable polymers used for suture production with potential application in suture fabrication by HME.

### 2.6. Material Requirements for HME

Suture threads are designed to meet many different patients’ needs. For instance, sutures for abdominal surgery are designed differently from sutures used in cataract surgery. Since no one type of suture is ideal for every surgical operation or suturing procedure, sutures are designed to have varying qualities depending on the area of utilisation [9,75]. Moreover, the choice of suture material in wound management, to a larger extent, depends on several factors that are broadly divided into: (i) patient-related and (ii) suture-related. For the former, age and medical condition of the patient, such as the number of tissue layers involved in wound closure, presence of oedema, and tension across the wound, need to be considered when developing the suture. While for the suture-related, depth of suture placement, expected time of suture removal or absorption after serving its function, possession of adequate tensile strength, and ability to elicit minimal or no inflammatory reactions are considered [39]. The rate at which the suture degrades is important, not only along the length of the suture but also at the knot [9]. Sutures must also possess significant pliability and flexibility for better handling during suturing and should be easy to sterilise. In addition, it must possess good knot placement, high knot security, free from irritation, infectious substances, and carcinogens, bearing in mind the final cost of their manufacture, which proportionately translates to the healthcare cost to be incurred by the patients [39,76]. It is worth noting that different tissues have differing requirements for suture support during healing. Some need only a few days, e.g., muscle, subcutaneous tissue, and skin, while others require weeks or even months, e.g., tendon and fascia tissues [9]. Thus, different types of wounds require different types of suturing materials.

In the remits of drug-eluting sutures produced via HME, the intrinsic properties of the thermoplastic material, the thermal stability of loaded API and that of other formulation additives, and the properties of the API-polymer blend during and after thermal exposure must be carefully studied. Thus, processing temperature, rheological properties, mechanical properties (such as tensile strength), and whether biocompatible and/or biodegradable are some of the critical quality attributes to consider when selecting a drug and/or a thermoplastic material for drug-eluting suture fabrication using HME. A selection tree highlights these factors in Figure 3. Although ambient temperature and relative humidity have been shown to significantly influence material behaviours during and after the HME process [60], the intrinsic property of the material is more critical since the environmental condition can be controlled [52].

Processing temperature: The heat deflection temperature at which materials deform plastically under load bends, whilst the continuous use temperature indicates the temperature at which materials can work continuously for an unlimited time [52]. The former is related to a lag time of heating necessary to initiate liquefication of the drug–polymer blend before extrusion, whilst the latter is related to the temperature maintained during the extrusion process. These temperatures are crucial to the choice of the HME material because, often, the mechanical resistance of the extrudates is a function of the operational temperature. Thus, it is important to select a material that is stable after exposure to these thermal conditions. During the extrusion process, the HME products often undergo shrinkage upon cooling, which is affected by the thermal properties of the material, such as its thermal conductivity. Uneven shrinkage often happens because of thermal gradients between the successive extrudates, which may lead to cracking of the extrudates [52,77,78].

Rheological properties: The fusion of the interface allows intralayer or interlayer adhesion between two extrudates to occur via a time-dependent mechanism involving polymer chain diffusion and viscous flow of materials. Rheological properties of the polymer and/or drug/polymer matrix, such as g viscosity and surface tension, influence the fusion of materials during HME [79,80]. Moreover, it has also been reported that even the distribution of a suitable filler in the polymer matrix can alter the rheological behaviours of the extruded filament and improve a stable filament diameter, thus facilitating the processability and stability of the final extrudates [52,81,82].

Tensile strength: In HME, the suture filament should retain sufficient tensile strength to prevent breakage during use, such as knotting. On the other hand, it also should be flexible enough to enable coiling and uncoiling during production and utilisation, respectively [77]. Thus, an ideal suture filament should have a minimum tensile strength of ∼400 Mpa [57,83].

Biocompatibility and biodegradability: For drug-eluting sutures fabricated through HME, the polymeric material should be biocompatible and/or biodegradable since most of them are designed as absorbable sutures. Depending on the area of use in the body, both attributes are required in most cases. Biocompatible thermostable polymers have the capacity to release drug(s) from drug delivery systems, such as drug-eluting sutures, without imparting significant changes in physiological functions or body fluids. On the other hand, biodegradable polymers degrade naturally in bioactive environments without releasing harmful by-products [84]. Moreover, often biodegradation of polymers results in the breakdown of long chains of polymers into monomers due to the enzymatic actions in the body [77,85]. Bio-based thermoplastic polymers are derived from renewable biomass sources such as starch, vegetable fats and oils, straw, and woodchips. It is, therefore, essential to explore different thermoplastic biobased materials for suture filaments to advance the range of naturally derived suture materials [52].

Other material requirements for HME include spinnability, smoothness, and aesthetics of the extrudates. The spinnability has to do with the reproducibility of the extrudate to be coiled after production and uncoiled during use and its consistent quality, while the smoothness and aesthetic appearance of the suture extrudates can significantly influence the suturing process and effective wound healing without scars [83,86]. Thus, these variables could directly or indirectly affect the suture production, its quality, and utilisation, the drug release, and/or the wound healing process.

Indeed, to increase traction and for a successful formulation of drug-eluting sutures by HME, thermoplastic polymers must be of pharmaceutical grade, the API should be miscible with the polymer and stable at extrusion temperature, and the extruded suture filaments should be pliable, flexible, with sufficient tensile strength and easy to manipulate during suturing. The thermoplastic polymer and API should ideally not require a very high temperature, and it is expected that the polymer melt temperature and the extrusion temperature should not be very close to or exceed the melting temperature of the loaded API to avoid degradation of the loaded drug(s) [58]. In other words, the selected API should not degrade at the melting temperature of the polymer [2]. Therefore, current efforts are centred on developing suture materials that possess these desired features along with additional capabilities, such as the potential to deliver drugs and cells to facilitate and/or augment wound healing [39].

## 3. Critical Quality Attributes of Drug-Eluting Sutures and Their Assessment

Certain suture characteristics are important for enhancing suture handling and wound healing [42,76]. This section discusses some of the critical quality attributes of drug-eluting sutures, including those prepared by HME, and their assessment procedures considered to be imperative during both suture prototyping and in-process control during scalability.

Suture size or cross-sectional diameter: Suture cross-sectional diameter is determined on individual suture strands, and there must be consistency in the value measured. Hence, the average value measured must be within the tolerances prescribed in the relevant pharmacopoeias. According to the USP, at least ten individual strands must be measured, and the measurement of each suture strand should be done at three points corresponding to one-fourth, one-half, and three-fourths of its length, depending on the suture type [77]. The measurement can be done using a suitable micrometre gauge, digital micrometre, or light microscopy as described by Parikh and colleagues, manipulated with the aid of ImageJ software (US National Institutes of Health, Bethesda, Maryland, https://imagej.nih.gov/ij/), [87]. The selection of suture size and cross-sectional diameter for wound closure is such that sutures should have the minimum cross-sectional diameter necessary for coaptation of the wound edges [87].

Surface morphology: A scanning electron microscope (SEM) with suitable magnification is used to study the surface morphology and cross-sectional characteristics of sutures before and after friction between suture and tissue or between suture and suture [8]. The surface morphology assessment can reveal information about the smoothness of the suture surfaces, internal diameter (size), suture shape, structure, and often how dispersed the API is within the suture polymer matrix. These pieces of information are essential in defining the physicochemical properties of the suture [88].

Mechanical properties (Tensile strength TS)*:* TS is the measure of maximum force that the suture withstands before it breaks [89]. It is the ratio between the maximum load the suture strand can withstand and the original cross-sectional area of the suture specimen. Therefore, TS is measured in units of force per unit area [90]. Indeed, one of the problems associated with drug-eluting sutures is low TS [4,10]. Even though absorbable sutures are expected to lose 50% of their TS within 60 days of their application [39], suture material should have and maintain adequate TS for its specified purpose. Sufficient TS of suture material is critically important for proper coaptation of flaps until wound healing is completed [91]. In most cases, suture materials lose between 70 and 80% of their initial TS [92], and that can lead to suture failure and disruption of the healing process. Thus, it is necessary to assess and guarantee stable suture TS to avoid breakage of the suture material [93]. Therefore, a deficit in the suture TS can lead to untimely rupture of the suture that can negatively affect the wound healing process.

TS test is usually conducted using a texture/mechanical testing machine using a set-up shown in Figure 4a. The movable clamp of the machine is attached from the top (upper clamp) with a defined force to be applied during the test, and the suture thread is attached to this clamp from the top; the lower head of the thread is held in the static clamp of the machine (lower clamp) [90,91,94,95]. The interpretation of such test results is generated from plots of load versus extension or stress versus strain curves (Figure 4b). From the plot, the following parameters can also be derived:Failure load is the load in Newton (N) at which the integrity of the suture thread is lost.Elongation at break (E%): Because of stress applied to suture material during the TS test, the dimension of the suture is stretched, resulting in strain generation. Percent elongation at break is determined by dividing the extension at the moment of suture break (L) by the initial gauge length of the specimen (L_0_) and multiplying by 100 according to Equation (1):
(1)$$E\%=\frac{L-L0}{L0}\times 100$$

Young’s modulus is calculated as the slope of the linear portion of the stress–strain curve. It is the measure of suture stiffness. Hard and brittle sutures are likely to have high Young’s modulus values and high TS.

Memory: The term memory in this context is closely related to elasticity and plasticity. It refers to a suture’s ability to assume a stable linear configuration after removal and stretching from packaging [96]. When a suture is free from irregular contortions and/or curling at the point of use that may interfere with surgical handling, such suture is said to have memory. However, sutures with significant memory are not flexible, which makes them difficult to manipulate during surgical handling, and significant memory may often necessitate additional knots [76,90,97]. This must, therefore, be remembered during wound closure and knot tying [18,90]. For drug-eluting sutures prepared by HME and in particular by generating a monofilament suture, there is a tendency to produce sutures with high stiffness because of two factors: First, due to thermal treatment of the drug–polymer blend, and second, due to the addition of API(s) which could alter the pliability of the suture thread making it difficult to handle. To address these challenges, the addition of a plasticiser could be explored to lower the stiffness while maintaining a good memory.

Coefficient of friction: This parameter describes how easily a suture thread passes through tissue during suturing [76,98]. Frictional properties of surgical sutures are important factors for consideration during suture production and utilisation because improper frictional properties of a suture can potentially cause damage to the tissue [99]. Indeed, elevated friction force in the suturing process may cause inflammation and pain to the patient, leading to a longer healing time and the secondary trauma of soft and/or fragile tissues [100]. The coefficient of friction is usually evaluated using a penetration friction apparatus and a linear elastic model using a texture analyser, as described by Zhang and co-workers. To measure the coefficient of friction, artificial skin is fixed in the gripper mount on the texture analyser connected to a force-measuring sensor. The tip of a surgical suture is left free, and the other suture end penetrates the skin using needle and is fixed onto the force-measuring sensor [99]. Chen and colleagues studied the suture-to-tissue friction force and suture-to-suture friction force using XF-1A surgical suture friction tester by measuring the static and dynamic resistance [8,101].

*Knot security*: This is the quality of a suture that allows it to be tied to the wound site securely with a few throws per knot [15,89]. Sutures with optimum knot strength abate the risk of wound dehiscence [8,76,102]. A knot stays connected to the wound site after suture because of the friction produced by one part of the knot acting on another, which relates to the coefficient of friction of the suture material. When a suture possesses a high coefficient of friction, it will exhibit good knot security but tends to scratch and drag through tissue during suturing [103,104].

An ideal knot should hold securely without fraying or cutting. For safety, a knot should have at least three throws with 3 mm long ends (Figure 5) [76]. Smooth surfaces decrease knot security, and, in such cases, there must be extra throws to compensate and assure knot security [76]. In other words, knot security depends on several factors, including the number of throws, suture technique, and the suture material [105,106]. Knot security can be assessed using a mechanical knotting machine, for instance, different surgical knot techniques can be performed using knot testing apparatus with a haemostat secured to one side and the other haemostat secured to a force gauge. The suture is then clipped between the two haemostats. From this setup, a standard knot-tying force is measured, from which the knot security satisfaction can be measured using a reference standard as described by Silver and his team [105].

Drug release: In vitro drug release from drug-eluting sutures is assessed by in vitro dissolution testing using test tubes in a heating water bath maintained at near-normal body temperature as possible, 37 ± 2 °C under stirring [38]. This test measures the rate and extent to which the drug is released from the suture. The dissolution media is defined by the nature of the drug(s) loaded, but in most cases, phosphate buffer pH 7.4 [38] or deionised water at room temperature [107] are used. At different time intervals often may take several days, samples are taken and analysed by UV-VIS spectroscopy or using validated high-performance liquid chromatography (HPLC) after building a suitable calibration curve of the loaded drug sample. However, there seems to be no rule of thumb to follow in conducting drug release tests for drug-eluting sutures, and internal conditions that are reproducible and reliable are used. For instance, Chen and colleagues tested the release properties of levofloxacin hydrochloride from antibacterial sutures using centrifuge tubes at 37 °C in aqueous media at different pH 6.3, 6.8, and 7.7 to mimic the skin pH values under normal and diseased conditions [108]. Thus, the test to use depends on the physicochemical properties of the loaded API and/or the physicochemical properties of the drug–polymer blend because, often, the type of polymer used also plays a pivotal role in the release properties of the loaded APIs.

Biodegradation profile: When a polymeric object-like suture is implanted in the body, a foreign body immune response is triggered [109,110]. The immune cells flock to the site of implantation to detect, quarantine, and potentially try to remove the implanted object [111]. These cells include macrophages, neutrophils, and fibroblasts, that secrete different types of enzymes, such as protease, lactate dehydrogenase, and acid phosphatase, which facilitate the suture degradation for absorbable sutures [109,112,113]. Thus, the biodegradation profile of drug-eluting sutures is evaluated in the presence and/or absence of proteolytic or hydrolytic enzymes that are commonly found in the human biological system [114]. Moreover, in the physiological environment, some biodegradable polymers are expected to degrade by hydrolysis of the ester bonds. In any of the above, the evaluation of the in vitro biodegradation of the sutures is made by measuring the weight loss of the experimental suture samples over time. A defined weight of the suture specimen is used and individually immersed in a known quantity of suitable medium and stored under controlled conditions. After each pre-determined degradation time interval, samples are taken, rinse with deionised or distilled water, and dry under ambient conditions. The percentage weight loss (*WL%*) of each specimen is then calculated using Equation (2) from n number of specimens [114,115].
(2)$$WL\%=\frac{Wi-Wf}{Wi}\times 100$$
where Wi is the initial dry weight, and Wf represents the dry weight after degradation and drying period. However, more complex modelling and simulation techniques can be used to predict and estimate polymer degradation and drug release dynamics from drug-eluting sutures as described by the work of Casalini and his team [116].

Antimicrobial effectiveness: In this assessment, two tests are usually conducted: (i) the qualitative assay to test the sensitivity of the antibacterial agent from the suture by assessing the zone of inhibition [117] and (ii) the sustained efficacy assay also known as serial plate transfer test. The latter test is performed to determine the duration of antibacterial activity of the drug-eluting sutures [44,114]. For compliance with the regulatory provisions when conducting these tests, reference is made to specific monographs in the relevant pharmacopoeias on the test to be conducted based on the type of API(s) loaded in the suture and the kind of microorganisms targeted.

Ex-vivo permeation studies: The Franz diffusion cells experiment is the most widely used technique to determine the drug permeation through the skin. It is considered the gold standard for assessing the delivery of drugs from a transdermal system [118]. However, for drug-eluting sutures, the ex-vivo permeation test aims to assess the rate and extent to which the loaded drugs penetrate beyond their target site of action, especially when the drug is expected to exert its effect locally. Since most drug-eluting sutures are designed to provide a maximum effective concentration of the drug within the vicinity of the wound and minimise systemic drug exposure, permeation study enables formulation scientists to understand the potential systemic disposition of loaded APIs in drug-eluting sutures and to develop strategies for minimizing it. It is worth to remember, that the ultimate goal of designing drug-eluting sutures is to control systemic exposure of the loaded drug thereby achieving its optimal concentration at the wound site to elicit the desired therapeutic effect(s).

Preferably, a permeation test is carried out using a specimen of human skin. However, ethical and economic dilemmas pose major constraints to the availability and use of human skin for permeation studies. Thus, isolated animal skin samples, such as those from rodents (guinea pig, rat, and mouse), rabbit, porcine, and primates, have found use as alternatives to human skin. These alternatives can be obtained easily and excised freshly at the point of use with nearly intact viability and minimal variability [119,120,121]. Porcine skin is the most preferred due to its structural resemblance to human skin, thus, it is routinely preferred for transdermal drug delivery systems evaluations, such as drug-eluting sutures [122,123,124,125,126].

The European Medicine Agency (EMA) recommends the use of six or more replicas from at least two or more skin donors in such permeation studies in order to minimise variability. When biological skin is used, the species’ body part should be stated. Storage condition and thickness of skin used should also be justified. The skin integrity should be confirmed and justified [118]. Permeation membrane (surface area) from the Franz diffusion cell in the range of 0.5 to 2 cm^2^ is acceptable (Figure 6). The receptor medium should be an aqueous buffer for water-soluble drugs, whilst hydroalcoholic media or aqueous buffers modified with solubility enhancers, such as surfactants, are preferred for poorly water-soluble drugs. The media composition in the receptor compartment should not affect the skin or membrane integrity. Therefore, surfactants in the receptor medium are limited due to potential interference with skin [118,127].

Samples are withdrawn from the receptor compartment at pre-set time intervals. The number of sampling points should be five or more so that an absorption profile of the drug substance can be obtained. For drug substances that permeate slowly, prolonged exposure times may be necessary. The medium in the receptor compartment should be agitated continuously to facilitate proper mixing. Samples can be analysed using a validated analytical technique, such as liquid chromatography-mass spectrometry (LC-MS) or HPLC [61,118]. To quantify the drug permeability, Fick’s law of diffusion, which describes the permeation process, is used. The flux and apparent permeability are calculated by using Equation (3):(3)$$Permeability(Papp)=\frac{Flux}{Cd}$$
where, Cd represents the initial drug concentration in the donor compartment [128]. Flux (J) is calculated by dividing the slope obtained by plotting the cumulative amount of drug permeated (M) through the skin against time (t) with the cross-sectional area of the membrane (A) exposed to the drug [123,129,130]. Despite the relevance of ex-vivo permeation assessment for drug-eluting sutures as discussed above to ascertain their effectiveness, there is a paucity of literature on experimental studies that evaluate the ex-vivo or in-vivo permeation properties of drug-eluting sutures. Therefore, it is the opinion of the authors that such studies should be integrated into the quality assessment of drug-eluting sutures.

Biocompatibility test: The inflammatory response promoted by the presence of foreign substances in the body within the wound triggers tissue reactivity. In most cases, the more suture material implanted, the greater the risk of tissue reaction. This response peaks within 2 to 7 days and is a function of the type and configuration of the suture as well [89]. The human keratinocyte cell line is commonly used to evaluate the cytotoxicity of drug-eluting sutures [10].

## 4. Future Perspectives

The function and efficacy of drug-eluting sutures depend largely on their physical and mechanical qualities. These sutures should have better handling properties, with all the biocompatibility requirements, and be free of allergens. To meet these specifications, it is necessary to subject the sutures to detailed in vitro quality assessments and pre-clinical trials and to evaluate their safety and efficacy in human trials [4]. Unfortunately, from the available literature, it appears that there are no harmonised quality passement protocols to evaluate the qualities of drug-eluting sutures with absolute certainty. Some of the assessment procedures are still evolving, and the methods of the suture production have also been budding recently, for example, the in vitro dissolution (drug release) assessment, unlike the case for conventional solid and semi-solid dosage forms where there is a clear explanation of the type of equipment and methods to be used in the different pharmacopoeias for in vitro dissolution tests. In contrast, there is no gold standard in vitro drug release method(s) described for drug-eluting sutures as some scientists reported using test tubes in a heating water bath while others made use of centrifuge tubes to conduct the test. Moreover, there is no explicit explanation about permeation studies for drug-eluting sutures in all the relevant pharmacopoeias as prescribed for other transdermal drug delivery systems. We are, therefore, of the opinion that the medicine regulatory agencies, especially FDA and EMA, should deeply review the quality assessment procedures of these medical devices to revise the existing ones where appropriate and generate new protocols where they do not exist. These agencies should consider the new trends and novelty of some of these sutures, such as the drug-eluting ones, and the smart sutures among emerging types. This will go along with ensuring the safety of these materials to the end-users.

Among the drug-eluting suture manufacturing processes that add the API during the manufacturing process, hot-melt extrusion appears to have high traction for future exploration since no solvent and no subsequent solvent removal step is required. Drug loading up to 20–30% weight by weight can be achieved by HME [2,38]. However, care must be taken because higher drug content lowers the tensile strength of the suture, which compromises an important quality attribute of the suture. Indeed, excess of the API can easily modify the suture properties, such as crystallinity and mechanical properties. Thus, formulation experts should ensure that the drug-eluting suture properties conform to the requirements of the relevant pharmacopoeias in terms of tensile strength. Additionally, HME can be used to tune the drug release profile in drug-eluting sutures by modifying the processing parameters and/or the composition of the drug–polymer blends (e.g., the use of copolymers), the suture morphology (e.g., thickness, porosity), and its architecture (e.g., core-shell dimensions) [2]. Another intriguing aspect is the fact that the new trends in the manufacturing processes of drug-eluting sutures have merely been motivated by the control of the drug loading and its release at the expense of the mechanical properties. From the technological point of view, assessment of the mechanical properties of drug-eluting sutures should go beyond just tensile strength measurement but should also go deeper to evaluate other critical suture mechanical parameters such as elongation at break, Young’s modulus, knot-pull tensile strength, and the coefficient of friction as they all provide important information relevant to understanding the suture quality and to enable comparison to pharmacopeial standards [2]

Finally, additive manufacturing (AM) has been shown to revolutionise the pharmaceutical manufacturing sector, especially pharmaceutical engineering of different drug delivery systems [52]. Indeed, HME has been demonstrated to be compatible with many of these AM technologies, such as fused deposition modelling three-dimensional (FDM 3D) printing [131,132,133], hot-melt pneumatic printing [134], and hot-melt ram extrusion printing [60]. Thus, coupling AM with HME to produce personalised drug-eluting sutures is an ambitious possibility that can be explored to leverage additional advantages that the AM has brought to the pharmaceutical field in dose personalisation and point-of-care manufacturing of drug delivery systems where drug-eluting sutures could also benefit from. Indeed, by this coupling technique, personalised drug-eluting sutures could be fabricated at the point of use or on-demand with the desired dimensions, such as thickness and length to control the drug release pattern. Of course, to achieve this, there must be a serious collaboration between the formulation experts, clinicians, and surgeons alike and medicine regulators with considerable advocacy.

## 5. Conclusions

Nowadays, there is increased traction towards the use of biodegradable synthetic polymers in the fabrication of suture threads owing to the advantages they offer being degraded within the human body through proteolytic or hydrolytic degradation without the need for suture removal post-surgery. Undoubtedly, the recent advancement in the fabrication process of drug-eluting sutures, such as the use of a hot-melt extrusion process employing biodegradable synthetic and/or semisynthetic polymers combined with APIs addition during manufacturing, is the most ideal development in the suture fabrication paradigm. HME is a one-step continuous process, easy to adopt and adapt at both prototyping and industrial scale. Indeed, the process does not require expensive equipment, which reduces cost and allows mass production. HME process should be the method of choice when the drug and polymer meet the required conditions for thermal treatment. There is no need for solvent addition, therefore, no residual solvent removal is needed, and there is a high chance of enhancing the solubility of poorly water-soluble APIs by HME. However, thermolabile APIs are not tolerant to melting temperatures, therefore, other production approaches should be sought for thermosensitive APIs such as electrospinning. Furthermore, from the therapeutic point of view, HME can be utilised in fabricating drug-eluting sutures loaded with antimicrobial agents, anti-inflammatory agents, and other therapeutics. This unique advantage can significantly reduce the chance of surgical site infections, alleviate pain and inflammatory response at the surgery site, and overall facilitate wound healing and reduced hospital stay.

## Figures and Tables

**Figure 1 materials-16-07245-f001:**
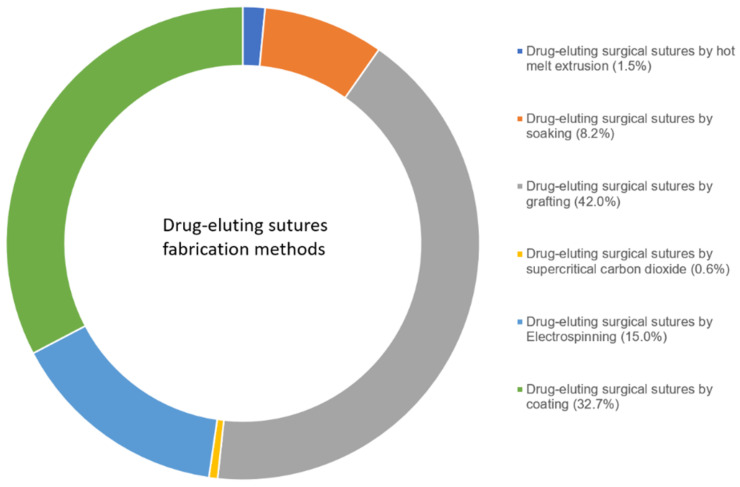
Summary of drug-eluting suture fabrication methods by research articles from 2013 to 2023 from the Scopus database as of 4 November 2023.

**Figure 2 materials-16-07245-f002:**
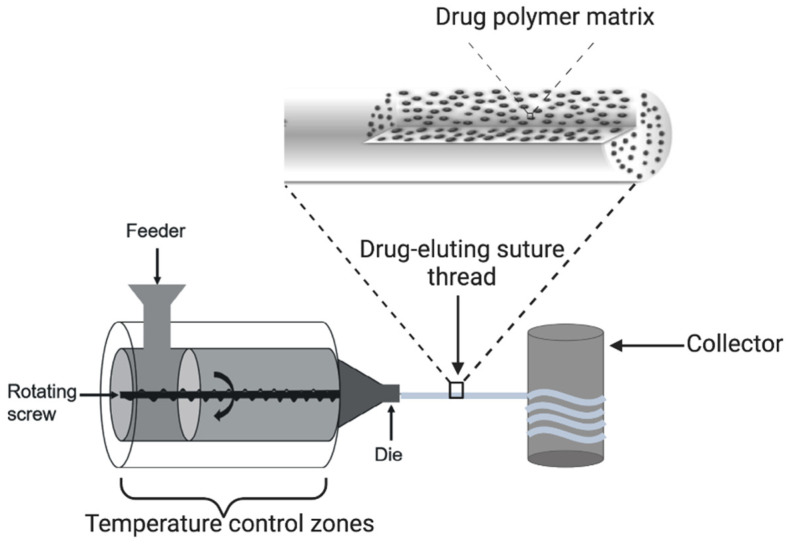
Schematic illustration of a hot-melt extrusion manufacturing process.

**Figure 3 materials-16-07245-f003:**
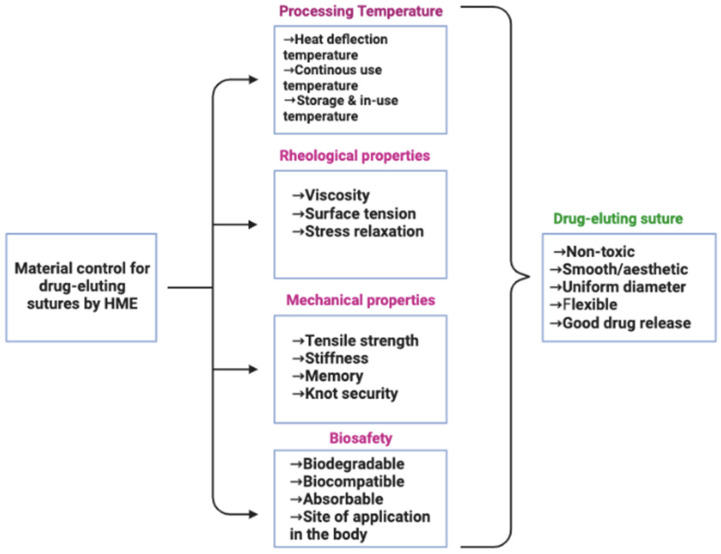
Schematic illustration of material control for drug-eluting suture production via HME.

**Figure 4 materials-16-07245-f004:**
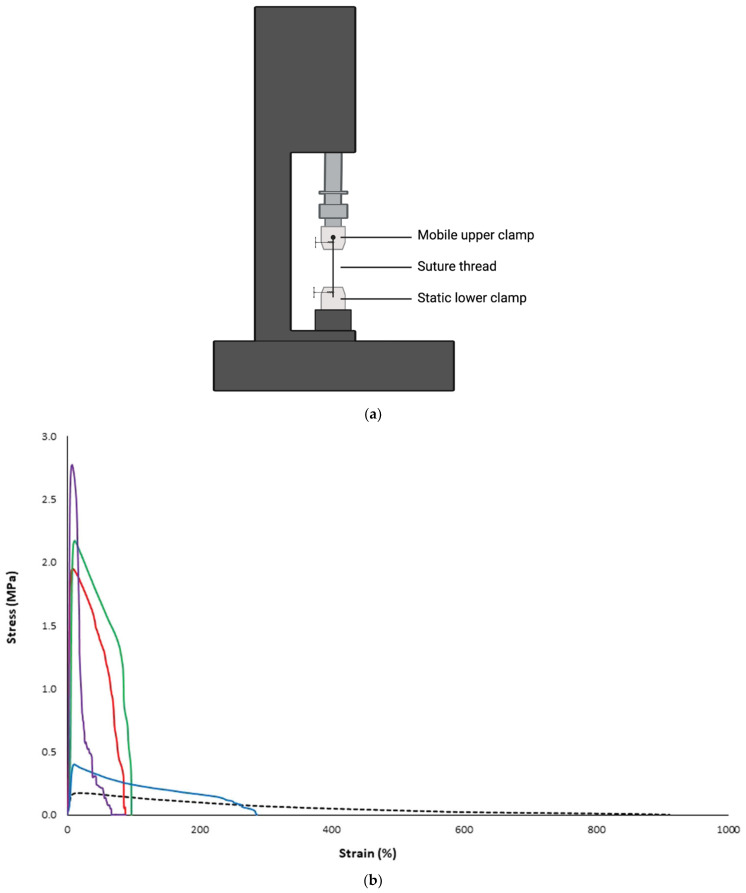
(**a**) Schematic illustration of suture mechanical properties measurement device. (**b**) Stress versus strain curves for drug-free filament (black dotted curve) and drug-loaded filaments (coloured solid curves). The stress generated from the plot represents the tensile strength, while the strain measures the percentage elongation at break, adopted from [58] with permission.

**Figure 5 materials-16-07245-f005:**
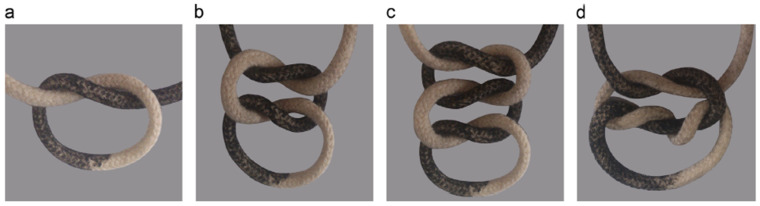
Hypothetical surgical knot configurations (**a**) simple knot, (**b**) square knot, (**c**) three throws knot (flat square), and (**d**) surgeon’s knot, adopted from [8] with permission.

**Figure 6 materials-16-07245-f006:**
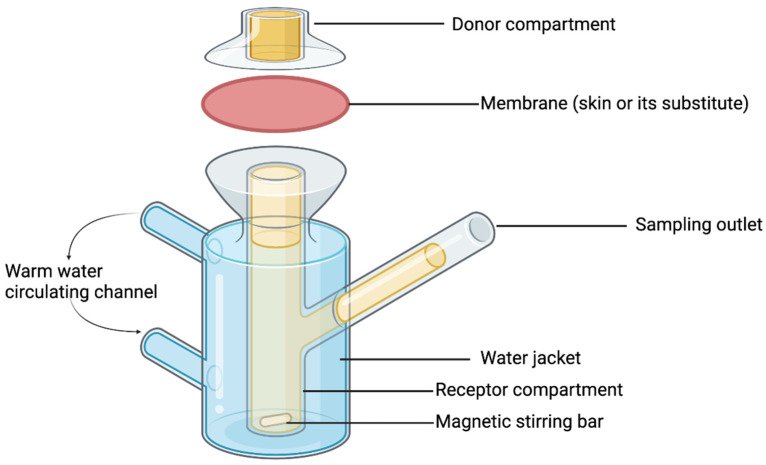
Schematic illustration of a vertical static Franz diffusion cell for ex-vivo penetration studies.

**Table 1 materials-16-07245-t001:** Examples of drug-eluting sutures produced by hot-melt extrusion.

HME Apparatus and Condition	Model Drug(s)	Suture Polymer/Excipients	Suture Structure	Suture Diameter (µm)	Special Features and Advantage	References
Extrusion was performed at 63 ± 1 °C for 5–20 min while mixing and extruded using a hot-melt extruder (CSI LE-075, PA, USA).	Diclofenac potassium	Polyethylene glycol-Poly(ε-caprolactone-chitosan-keratin blend	Monofilament	500–700	Amorphous and miscible solid dispersions were created. Rapid and sustained drug release rates were achieved with the PEG/PCL/chitosan/keratin blends at various combinations. Presence of hydrophilic and phobic polymers improved the solubility of the diclofenac potassium with a tunable release rate.	[38]
HAAKE twin screw extruder using a screw speed of 20 rpm and applying a temperature profile going from 60 °C, at the feed zone, to 100 °C at the die zone	Nanohybrid ofhydrotalcite andDiclofenac (HT-Dic)	Poly(ε-caprolactone) (PCL)	Monofilament	300	Controlled release of the drugs and complete release after 55 days. In vivo results show a reduction of inflammatory responses associated with drug-loaded suture	[57]
Counter-rotating twin-screw compounder (Brabender, D045 mm, L/D07) with a thermal profile of 40–50–70–100 °C and a speed of 64 rpm.	Chlorhexidinediacetate (CHX)	Poly(caprolactone) (PCL)	Monofilament	300 ± 10	An initial rapid release followed by a slow-release pattern The thermal profile condition used at 100 °C does not compromise the antibacterial activity of Chlorhexidine diacetate No toxic effect from the suture to compromise the cell viability of human fibroblasts in vitro.	[56]
Dynisco extruder hopper and sutures were extruded from a melted mixture of PLGA pellets and lyophilised CpG ODN at about 70 °C	Cytosine–phosphorothioate–guanineoligonucleotides (CpG ODN)	Polylactic acid-co-glycolic acid (PLGA)	Monofilament	600	Sustained release of CpG ODN over 35 days Suppression of neuroblastoma recurrence in mouse	[68]
The polymer was melt-spun at 180 °C under nitrogen, allowed to heat up for 10 min, and fibres drawn at 192 °C	Nitric oxide (NO)	Acrylonitrile-co-1-vinylimida-zole (AN/VIM) copolymer.polycaprolactone (PCL) used as a secondary coating	Monofilament	0.366	The addition of polycaprolactone coating delayed the release of NO from the sutures. The additional coatings in the suture prolonged NO delivery to a wound and enhanced wound healing.	[69]
Drug polymer melt was extruded in a Collin Zk25 twin-screw extruder using a temperature profile along the extruder with six heater zones set at 70, 80, 80, 75, 70, and 60 °C from the feed to the die end and a screw speed of 60 rpm.	Ibuprofen	Poly(ε-caprolactone) (PCL)	Monofilament	Not provided	Ibuprofen molecules were well dispersed and evenly distributed within the PCL matrix. The release of ibuprofen from PCL was retarded when both Cloisite 20A and Somasif MEE surfactants were dispersed in the PCL-ibuprofen blend.	[66]
HAAKE MiniLab micro compounder (Thermo-Haake, Karlsruhe, Germany) at 155 °C and screw speed of 20 rpm	Carvedilol	Eudragit^®^ E	Monofilament	20	Up to 20% of carvedilol was loaded Fast release of carvedilol, which has poor water solubility. Comparable drug loading and drug release with suture fibres with similar compositions produced by solvent-free melt electrospinning and solvent-based electrospinning.	[55]

## Data Availability

The data presented in this study are available on request from the corresponding author.

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
