# Peer review of "Drug-Eluting Sutures by Hot-Melt Extrusion: Current Trends and Future Potentials"

_materials, 2023, doi:10.3390/ma16227245_

Round 1
Reviewer 1 Report
Comments and Suggestions for Authors
The authors discuss an interesting topic and do a good job writing comprehensive review about use of HME for developing drug eluting sutures. However, the review needs more discussion regarding the limitations and have a rationale for using HME over other advanced bio fabrication techniques which put the practicality of preferring HME over other biofabrication strategies into question. I recommend that the authors address these comments in detail and include discussion about these points in the manuscript so the manuscript can be considered for publication post a secondary review:
1) The authors suggest use of hot melt extrusion techniques to produce drug eluting scaffolds; what is the effect of the temperature on the half life of the intended drug? Have any studies discussed this as a potential limiting factor? is the drug added post processing?
2) It is possible to more efficiently electrospin and generate co-axially electrospun fine filamentous sutures which can allow us to generate drug encapsulated sutures without potentiay affectying the drug's half life, why is use of HME better or more preferred? this should be discussed.
3) As a follow up a table comparing the efficacy of HME fabricated drug-eluting sutures with other drug-eluting sutures manufactured using other biofabrication techniques should be added to this review.
A detailed discussion on these points is needed before this paper can be reveiwed again and then considered for publication.
Author Response
The authors suggest use of hot melt extrusion techniques to produce drug eluting scaffolds; what is the effect of the temperature on the half life of the intended drug? Have any studies discussed this as a potential limiting factor?
Response
A critical discussion on the limitation of HME on drug is now included as a chapter (2.4) in the revision:
2.4 Limitations of HME for the production of drug-eluting sutures on drug half-life
“Limitations of HME for the production of drug-eluting sutures on drug half-life is among the most critical limitations of HME is the thermal degradations of the drug and/or drug-polymer mix. HME possesses intrinsic limitations with respect to processing temperature on thermolabile active pharmaceutical ingredients on the drug half-life due to high thermal and mechanical energy applied to the mix during extrusion [71]. However, a study by Deng et al., indicates that mixing an appropriate amount of drug and polymer(s) can result in a blend with suitable thermal stability [38]. For instance, addition of a 4% weight of chitosan/keratin blend into poly(ε-caprolactone) and Poly(ethylene glycol) blend, the thermal stability of the suture fibers loaded with diclofenac potassium was maintained at a degradation temperature of 378 ËšC similar to pure poly(ε-caprolactone) of 375 ËšC. Thanks to the solvent-free manufacturing process by HME. Similarly, sometimes the degradation products from the HME process need to be investigated and monitored to ensure their limits do not exceed the recommended amount in pharmaceutical products. As an example, Khalid and colleagues found an impurity in the diclofenac sodium as a degradation product from orodispersible films manufactured by HME 3D printing due to thermal exposure of the drug. The detected amount monitored under a 3-month accelerated stability study was found to be below the acceptable limits set by the European Pharmacopoeia (Ph. Eur.) [55]. Thus, in a broader perspective, the detection of drug degradation products does not mean a complete stoppage from utilizing the HME process, but additional quantification and follow-up analyses are needed to justify the adoption of HME or otherwise. EHMTherefore, as a rule of thumb, a preliminary assessment should be conducted on the individual products and the drug-polymer mix during the pre-formulation phase. Traditionally, chromatographic techniques such as high-performance liquid chromatography (HPLC) are used to evaluate the chemical stability of mixed components subjected to high temperature and shear stresses during HME. HPLC can be equipped with different detectors, such as mass spectrometry (MS) because it enables the identification of degradation products [72]. Other chromatographic techniques attached to HPLC used to assess degradation profiles from HME products include high-performance liquid chromatography Photo Diode Array (HPLC-PDA), Gas-Chromatography Mass Spectrometry (GC-MS), and Gel Permeation Chromatography (GPC) [71, 73].
In addition to chromatographic techniques that are essentially used for the identification and quantification of drugs and their degradation products from HME, evaluation of the solid physical state is equally important in relation to the stability of extrudates, since HME process frequently promotes the disruption of the crystal lattice and the recovery of an amorphous or partially amorphous solid. This in turn influences stability during the product shelf-life, crystallization tendency, drug dissolution, and drug bioavailability [71]. As such, to understand and follow modifications at the solid state of a drug-polymer system under the HME process and also to evaluate the physicochemical stability of the system during its shelf-life, various techniques are employed such as X-ray powder diffraction (XRPD), differential scanning calorimetry(DSC), modulated temperature DSC (MTDSC), hot stage microscopy (HSM) or hot stage polarized light microscopy (HS-PLM), and atomic force microscopy (AFM) are of prime importance [71, 74].
Other constraints of HME are related to the limited number of heat-stable polymers, the requirement of raw materials with high flow properties, limited physical stability, and often the precipitation of drugs during dissolution [73]. The later limitations aside the thermal degradation potentials can be circumvented when the drug-polymer establishes stable interactions, such, as hydrogen bonds, dipole-dipole, and Vander Waals interactions [73]"
Is the drug added post processing?
Response
No, in HME, one of the advantages is that the drug(s) are added alongside other excipients, mixed together and extruded as a single-step processing that avoids many other unit operations such as drying to remove residual solvents and solvent effects on the finished product. This is discussed in line 210
2) It is possible to more efficiently electrospin and generate co-axially electrospun fine filamentous sutures which can allow us to generate drug encapsulated sutures without potentiay affectying the drug's half life, why is use of HME better or more preferred? this should be discussed.
Response
Yes, as discussed in the manuscript from line 185 with the merits and demits of electrospinning both uniaxial and co-axial, such as poor mechanical properties due to the porous nature of the fibers. Moreover, the advantages of HME for the production of drug-eluting sutures over the electrospinning method have been extensively discussed such as the ease of continuous manufacturing and scalability, avoiding the use of solvent e.t.c.
3) As a follow up a table comparing the efficacy of HME fabricated drug-eluting sutures with other drug-eluting sutures manufactured using other biofabrication techniques should be added to this review.
Response
We thank the reviewer for this comment, however, this review focused on drug-eluting sutures by HME, and an intensive list of examples was provided in Table 1 citing several examples of other suture preparation techniques. Moreover, to address this important question by the reviewer, a line and new Figure 1 have been added in the manuscript to summarize the development of drug-eluting sutures by different techniques in the last decade as follows:
“Indeed, a literature search from the Scopus database with ‘drug loaded surgical sutures by hot melt extrusion’ and by other surgical sutures fabrication techniques as listed in Figure 1 as of November 4th 2023 returned very few numbers of research articles on sutures preparation by hot melt extrusion, a total of 38 (2% relative to other preparation methods) over the last ten years (2013 to 2023). The most frequently reported technique is grafting (42%), followed by coating technique (33%), then electrospinning (15%). This shows a significant paucity of research output on drug-eluting sutures by hot melt extrusion despite its promising potential.”

Figure 1. Summary of drug-eluting sutures fabrication methods by research articles from 2013 to 2023 from the Scopus database as of November 4th 2023

Reviewer 2 Report
Comments and Suggestions for Authors
In the current review paper, the authors have summarized the current trend of different types of sutures, especially the drug-eluting sutures and therefore the introductions of related production and materials. As a review paper, there are only few papers up to date (none of them has been published in 2023). All the covered topics are quite general and lack of deeper insights. There are so many format issues. I cannot recommend the publication of the paper in its current form.
1. It would be useful to provide a diagram plot/pie chart for the usage of different sutures, and the revolution footage as a function of year in terms of publications.
2. When listing the advantages of drug-eluting sutures and the later hot-melt extrusion in drug-eluting sutures, the citations should be properly placed and correctly introduced, instead of a single package of citations.
3. There are four figures in total, basically all of them are illustrations and cannot provide strong evidence to the point where the figure has been referred. The figures should be more meaningful, highlighting new developments and new discovers rather than simple illustrations.
4. The authors have mentioned that the Figure 3(b) is adopted from Ref. 55 (I assume also by the author), but the y-axis is completely different from what it looks like in the Ref. 55, could the authors explain this inconsistency? In addition, I can even see the editing traces of the y-axis.
5. The labeling of section 2 has some problems, I can only find section 2, 2.1, 2.1.2 and 2.1.3, I believe the latter two sections should be 2.2 and 2.3, the authors should really check the manuscript carefully before submitting it to the journal.
6. Additional space is found between words at multiple places, for example (but not limited to), Line 22, 77, 93, 95, 156, 202, 450, etc,.
7. Some paragraphs have indent in front while so many other paragraphs are not, this inconsistency makes me worried for the drafting process.
Comments on the Quality of English LanguageModerate editing of English language required
Author Response
In the current review paper, the authors have summarized the current trend of different types of sutures, especially the drug-eluting sutures and therefore the introductions of related production and materials. As a review paper, there are only few papers up to date (none of them has been published in 2023). All the covered topics are quite general and lack of deeper insights. There are so many format issues. I cannot recommend the publication of the paper in its current form.
1. It would be useful to provide a diagram plot/pie chart for the usage of different sutures, and the revolution footage as a function of year in terms of publications.
Response
We thank the reviewer for this comment. To address this important question by the reviewer, a line and new Figure 1 have been added in the manuscript as follows:
“Indeed, a literature search from the Scopus database with ‘drug loaded surgical sutures by hot melt extrusion’ and by other surgical sutures fabrication techniques as listed in Figure 1 as of November 4th 2023 returned very few numbers of research articles on sutures preparation by hot melt extrusion, a total of 38 (2% relative to other preparation methods) over the last ten years (2013 to 2023). The most frequently reported technique is grafting (42%), followed by coating technique (33%), then electrospinning (15%). This shows a significant paucity of research output on drug-eluting sutures by hot melt extrusion despite its promising potential.”

Figure 1. Summary of drug-eluting sutures fabrication methods by research articles from 2013 to 2023 from the Scopus database as of November 4th 2023
2. When listing the advantages of drug-eluting sutures and the later hot-melt extrusion in drug-eluting sutures, the citations should be properly placed and correctly introduced, instead of a single package of citations.
Response
We thank the reviewer for this comment, citations are now properly placed at the beginning of each of the mentioned paragraphs.
3. There are four figures in total, basically all of them are illustrations and cannot provide strong evidence to the point where the figure has been referred. The figures should be more meaningful, highlighting new developments and new discovers rather than simple illustrations.
Response
We thank the reviewer for this comment, the illustrations were aimed at helping the reader understand the principles of the hot melt extrusion technique and the critical quality attributes parameters of the extrudates. However, an additional Figure (Figure 1) has been added as suggested by the reviewer to highlight the recent trends in the drug-eluting suture fabrication methods. As follows:
Indeed, a literature search from the Scopus database with ‘drug loaded surgical sutures by hot melt extrusion’ and by other surgical sutures fabrication techniques as listed in Figure 1 as of November 4th 2023 returned very few numbers of research articles on sutures preparation by hot melt extrusion, a total of 38 (2% relative to other preparation methods) over the last ten years (2013 to 2023). The most frequently reported technique is grafting (42%), followed by coating technique (33%), then electrospinning (15%). This shows a significant paucity of research output on drug-eluting sutures by hot melt extrusion despite its promising potential.

Figure 1. Summary of drug-eluting sutures fabrication methods by research articles from 2013 to 2023 from the Scopus database as of November 4th 2023
4. The authors have mentioned that the Figure 3(b) is adopted from Ref. 55 (I assume also by the author), but the y-axis is completely different from what it looks like in the Ref. 55, could the authors explain this inconsistency? In addition, I can even see the editing traces of the y-axis.
Response
We thank the reviewer for this comment; the correct Figure has been inserted in the manuscript accordingly.

Figure 3b. Stress versus strain curves for drug-free filament (black dotted curve) and drug-loaded filaments (colored solid curves). The stress generated from the plot represents the tensile strength while the strain measures the percentage elongation at break, adopted from [55] with permission.
5. The labeling of section 2 has some problems, I can only find section 2, 2.1, 2.1.2 and 2.1.3, I believe the latter two sections should be 2.2 and 2.3, the authors should really check the manuscript carefully before submitting it to the journal.
Response
We thank the reviewer for this comment, the sections and sub-sections labeling have been corrected accordingly.
6. Additional space is found between words at multiple places, for example (but not limited to), Line 22, 77, 93, 95, 156, 202, 450, etc,.
Response
We thank the reviewer for this comment, the spaces as mentioned have been corrected accordingly.
7. Some paragraphs have indent in front while so many other paragraphs are not, this inconsistency makes me worried for the drafting process.
Response
We thank the reviewer for this comment, all paragraphs are now correctly indented according to the journal’s guidelines.

Reviewer 3 Report
Comments and Suggestions for Authors
Thank you for the draft. I enjoyed studying your publication. The publication addresses a topic that, to my knowledge, has surprisingly hardly been covered in relevant reviews recently. In my opinion, this review is very helpful. Your work is very well researched, structured and to the point on the chosen topic.
Author Response
The authors thank the reviewer for the comments
Reviewer 4 Report
Comments and Suggestions for Authors
This review comprehensively summarized the current status of drug-eluting sutures by hot-melt extrusion, and provided some future perspectives. The background information of drug-eluting suture fabrication has been fully introduced, and the advantages and disadvantages of hot-melt extrusion have been well listed and discussed. This manuscript could be accepted after minor revision. Please address the following comments.
1. Line 271, is 'Table 2' supposed to be 'Table 1'? Please fix this typo.
2. For the drawbacks of HME, the authors regarded the thermal compatibility of APIs at processing temperature of polymers as the main reason of HME not being widely applied in manufacturing. In the table, the authors listed several examples, and I'm wondering, in the current manufacturing of drug-eluting sutures, what is the approximate percentage of those polymers and APIs that have potential applications in suture fabrication using HME?
3. In 2.1.3, for the materials requirements for HME, is hydrophobicity/hydrophilicity also a critical property that affects the material behaviors?
4. In the current manufacturing, which techniques are applied for those poorly water-soluble drugs besides hot-melt extrusion and melt electrospinning?
Author Response
- Line 271, is 'Table 2' supposed to be 'Table 1'? Please fix this typo.
Response:
The typo has been corrected.
- For the drawbacks of HME, the authors regarded the thermal compatibility of APIs at processing temperature of polymers as the main reason of HME not being widely applied in manufacturing. In the table, the authors listed several examples, and I'm wondering, in the current manufacturing of drug-eluting sutures, what is the approximate percentage of those polymers and APIs that have potential applications in suture fabrication using HME?
Response:
A short statement regarding the statistics on the percentage of polymers and APIs used in drug-eluting sutures is now included in line 220:
“There are few studies that have reported the manufacturing of drug-eluting sutures by HME [65, 67]. This is largely due to the elevated processing temperature required for the extrusion. Thus, it cannot be applied to thermolabile drugs [1]. Polycaprolactone has received much attention for use in this regard because it is biodegradable and is approved by the FDA for human use [38, 66, 67].
- In 2.1.3, for the materials requirements for HME, is hydrophobicity/hydrophilicity also a critical property that affects the material behaviors?
Response
Yes, API candidates that exhibit hydrophobicity and hence present low solubility can be constructed as amorphous solid dispersion within the sutures. In this regard, an added advantage of improved solubility is of the API is achieved. This is mentioned earlier under 2.1.
- In the current manufacturing, which techniques are applied for those poorly water-soluble drugs besides hot-melt extrusion and melt electrospinning?
Response
The other methods used are captured under 2.1., line 179, however, another technique not mentioned in that line has now been included: “soaking”.
Furthermore, additional information regarding the current trend in drug-eluting suture production from published literature has been added in line 187 with additional Figure 1 as follows:
“Indeed, a literature search from the Scopus database with ‘drug loaded surgical sutures by hot melt extrusion’ and by other surgical sutures fabrication techniques as listed in Figure 1 as of November 4th 2023 returned very few numbers of research articles on sutures preparation by hot melt extrusion, a total of 38 (2% relative to other preparation methods) over the last ten years (2013 to 2023). The most frequently reported technique is grafting (42%), followed by coating technique (33%), then electrospinning (15%). This shows a significant paucity of research output on drug-eluting sutures by hot melt extrusion despite its promising potential.”

Figure 1. Summary of drug-eluting sutures fabrication methods by research articles from 2013 to 2023 from the Scopus database as of November 4th 2023

Round 2
Reviewer 2 Report
Comments and Suggestions for Authors
The authors have properly addressed all the concerns, thus, I don't have more questions this time.
Comments on the Quality of English LanguageMinor editing of English language required